# Overhydration Assessed Using Bioelectrical Impedance Vector Analysis Adversely Affects 90-Day Clinical Outcome among SARS-CoV2 Patients: A New Approach

**DOI:** 10.3390/nu14132726

**Published:** 2022-06-30

**Authors:** Isabel Cornejo-Pareja, Isabel M. Vegas-Aguilar, Henry Lukaski, Antonio Talluri, Diego Bellido-Guerrero, Francisco J. Tinahones, Jose Manuel García-Almeida

**Affiliations:** 1Instituto de Investigación Biomédica de Málaga (IBIMA), Virgen de la Victoria University Hospital, 29010 Málaga, Spain; 2Centro de Investigacion Biomedica en Red de la Fisiopatología de la Obesidad y Nutricion (CIBEROBN), Instituto de Salud Carlos III (ISCIII), 29010 Malaga, Spain; 3Department of Endocrinology and Nutrition, Virgen de la Victoria University Hospital, Malaga University, 29010 Malaga, Spain; isabel.mva13@gmail.com (I.M.V.-A.); jgarciaalmeida@gmail.com (J.M.G.-A.); 4Department of Kinesiology and Public Health Education, University of North Dakota, Grand Forks, ND 58202-7166, USA; henry.lukaski@und.edu; 5Antonio Talluri BME, Fatbyte, Inc., Bagno a Ripoli, 50012 Florence, Italy; tony@tonytalluri.com; 6Department of Endocrinology and Nutrition, Complejo Hospitalario Universitario de Ferrol, 15405 Ferrol, Spain; diegobellido@gmail.com

**Keywords:** hydration status, hydration fat-free mass, extracellular water, total body water, COVID-19, survival and mortality analysis

## Abstract

Background: COVID-19 has taken on pandemic proportions with growing interest in prognostic factors. Overhydration is a risk factor for mortality in several medical conditions with its role in COVID-19, assessed with bioelectrical impedance (BI), gaining research interest. COVID-19 affects hydration status. The aim was to determine the hydration predictive role on 90 d survival COVID-19 and to compare BI assessments with traditional measures of hydration. Methods: We studied 127 consecutive COVID-19 patients. Hydration status was estimated using a 50 kHz phase-sensitive BI and estimated, compared with clinical scores and laboratory markers to predict mortality. Results: Non-surviving COVID-19 patients had significantly higher hydration 85.2% (76.9–89.3) vs. 73.7% (73.2–82.1) and extracellular water/total body water (ECW/TBW) 0.67 (0.59–0.75) vs. 0.54 (0.48–0.61) (*p* = 0.001, respectively), compared to surviving. Patients in the highest hydration tertile had increased mortality (*p* = 0.012), Intensive Care Unit (ICU) admission (*p* = 0.027), COVID-19 SEIMC score (*p* = 0.003), and inflammation biomarkers [CRP/prealbumin (*p* = 0.011)]. Multivariate analysis revealed that hydration status was associated with increased mortality. HR was 2.967 (95%CI, 1.459–6.032, *p* < 0.001) for hydration and 2.528 (95%CI, 1.664–3.843, *p* < 0.001) for ECW/TBW, which were significantly greater than traditional measures: CRP/prealbumin 3.057(95%CI, 0.906–10.308, *p* = 0.072) or *BUN*/creatinine 1.861 (95%CI, 1.375–2.520, *p* < 0.001). Hydration > 76.15% or ECW/TBW > 0.58 were the cut-off values predicting COVID-19 mortality with 81.3% and 93.8% sensitivity and 64 and 67.6% specificity, respectively. Hydration status offers a sensitive and specific prognostic test at admission, compared to established poor prognosis parameters. Conclusions and Relevance: Overhydration, indicated as high hydration (>76.15%) and ECW/TBW (>0.58), were significant predictors of COVID-19 mortality. These findings suggest that hydration evaluation with 50 kHz phase-sensitive BI measurements should be routinely included in the clinical assessment of COVID-19 patients at hospital admission, to identify increased mortality risk patients and assist medical care.

## 1. Introduction

The coronavirus disease 2019 (COVID-19) has high human transmission and considerable mortality rate [1]. Thus, clinicians actively seek to identify physiological factors associated with increased risk of non-survival.

Researchers continue to identify prognostic factors for morbidity and mortality of SARS-CoV-2 [2]. Whereas investigations have been focusing on blood biochemical, pre-existing diseases or comorbidities, drug treatments, and basic clinical variables (O_2_ saturation, temperature, or heart rate), observational reports identify excess fluid accumulation, acral and pulmonary edema in SARS-CoV-2 patients [3,4]. Emerging evidence identifies bioelectrical impedance (BI) derived hydration status, notable overhydration at admission, as a significant predictor of non-survival in these patients [5,6]. Overhydration refers to an imbalance in fluid distribution between extracellular water (ECW), and intracellular water (ICW) volumes with the expansion of ECW associated with systemic inflammation and aggressive fluid administration [7]. Overhydration is a risk factor for mortality in several medical conditions, such as critical illness, heart, and kidney failure, and a significant mortality predictor (OR 4.38; 95%CI, 2.76–6.94) [8].

The accurate, rapid bedside hydration assessment in hospitalized patients is a challenge, since available methods are impractical and costly for point-of-care application, have limited precision for an individual, and lack individual-specific normal values [9]. Physical characteristics such as body weight, blood pressure, central venous pressure, presence/absence of edema, and laboratory analysis are non-specific to provide a reliable estimate of hydration for an individual patient. However, in COVID-19 patients, the validity of these methods is limited by disease characteristics and compliance with the rules for controlling the spread of infection [10].

Real-time assessment of hydration status alternatively utilizes bioelectrical impedance estimates and classification with bioelectrical impedance vector analysis (BIVA) measurements [11,12]. This method is bodyweight independent and relies on sample-specific multiple regression models to classify fluid volumes and distribution for an individual. The BI method relies on the conduction of an applied alternating current by water and electrolytes, and delay of the current by cell membranes and tissue interfaces to derive measurements of resistance (R), which is associated with total body water (TBW) volume, and series reactance (Xc), an index of body cell mass (BCM) relative density [11,13,14]. The geometric relationships of R and Xc demonstrate that the impedance (Z) vector (Z^2^ = R^2^ + Xc^2^) is inversely related to TBW [11], and the phase angle (PhA) [PhA = arc tan (Xc/R)•(Π•360°)] is inversely related to tracer dilution measurements of ECW/ICW [15] and cumulative fluid balance in critical patients [16].

This study aimed to determine whether hydration measurements derived using BI measurements and BIVA, hydration percentage, and ECW/TBW ratio, could predict mortality in a cohort of patients with COVID-19. This study incorporates global clinical scales [COVID-19 Spanish Society of Infectious Diseases and Clinical Microbiology (SEIMC) Score] [17] and other laboratory prognostic markers of inflammation and venous congestion [18,19]. It also compared the predictive value of BI-derived parameters with clinical biochemical indicators of hydration as mortality risk indicators in hospitalized COVID-19 patients.

## 2. Materials and Methods

### 2.1. Setting Study

In this single-center, longitudinal cohort study, we enrolled a sample of patients admitted consecutively to a hospital area care for COVID-19 in Infectious Disease Unit, in Virgen Victoria Hospital (Malaga, Spain), measured between 6–17 April 2020 followed by 90 d outcomes. All patients were diagnosed with COVID-19 pneumonia according to World Health Organization interim guidance [20] using real-time reverse transcriptase-polymerase chain reaction assays. This study was approved by the Ethics Committee of Virgen Victoria “PhA_COVID-19-201023” (Institutional human review board).

### 2.2. Measurements

We measured the hydration status of the patients within 72 h after hospital admission. BI measurements were obtained with a 50 kHz, phase-sensitive impedance analyzer (BIA101 Bioimpedance Vector Analyzer (AKERN, Pontassieve, Italy) that introduces 800 μA using the standard tetrapolar supine technique [12,21].

BIVA was performed [12] using the RXc graph to classify hydration status and body soft tissue mass. Resistance is inversely related to fluid content, indirectly reflecting soft tissue mass. The Bioimpedance Analysis (BIA) technique has high technical accurately (R = +/−1%, Xc = +/−2%) daily controlled by a test circuit yielding R = 383 Ω and Xc = 45 Ω, with a low technical error of measurement (0.07–0.30%) and a biological variation of measurement (1.9%) for repeated measurements in healthy adults [22,23].

The BIA instrument uses Hydragram^®^ software to provide an interpretative evaluation of dehydration or overhydration. The individual fat-free mass (FFM) hydration factor (TBW/FFM, %) is given as a numerical reference. Thus, changes in fluid balance can be recognized fast and reliably, assisting therapy follow-up, as body water volume changes are turned into an easy-to-understand hydration scale. Hydration state [24] is expressed as the individual hydration in percentage. Euhydration is described as 72.7–74.3% with overhydration exceeding 74.3% + 1 SD of euhydration and dehydration being less than 72.7% − 1 SD as derived by Moore and others [25,26].

Biagram^®^ illustrates the distribution of phase-sensitive 50 kHz BI direct measurements of Xc and PhA from the phase-sensitive BI analyzer for an individual [27]. The measurements are height and weight independent evaluation methods to detect the extracellular/intracellular space proportion in clinical practice. Individual points in different regions of the reference Biagram^®^ graph have specific interpretations of fluids distribution spaces. Points below the lower line of nomogram are related principally to extracellular conductive pathway due to ECW expansion and/or contraction of capacitive membranes, points above the upper line are associated predominantly with alterations of ICW mainly due to dehydration, and points between the lines generally indicate normal extracellular/Intracellular (E/I) proportion [28].

Hydration status is described also as the ECW/TBW ratio adjusted by age and gender [29]. For Male: $\frac{\mathrm{ECW}}{\mathrm{TBW}}ratio=\frac{40.02+0.0714\;*\;\mathrm{Age}}{100}$. For female: $\frac{\mathrm{ECW}}{\mathrm{TBW}}ratio=\frac{43.28+0.0470\;*\;\mathrm{Age}}{100}$. In a healthy state, ECW/TBW ratio should fall within the range 0.360–0.390 [30].

Bodyweight and standing height were determined at admission and before BI measurement; weight measurements were with a scale (100 g sensitivity); while height measurements were measured with a 2 mm sensitivity laser height rod. A standardized quality assurance protocol was used by trained healthcare professionals to reduce measurement variability. All BI measurements were obtained with the patient supine on a standard hospital bed, at least five to ten minutes in a supine position before registering BI values, in order to achieve an even fluid distribution from standing to recumbency. This is a well-known phenomenon affecting R and Xc values. Final R and Xc acceptance, only after substantial stabilization of BIA measurement values, is ±2 Ω for R and ±1 Ω for Xc.

### 2.3. Clinical and Analytical Variables

We determined the following clinical assessments: age, sex, any comorbidities, (e.g., history of diabetes, hypertension, dyslipidemia, obesity, heart disease, pulmonary disease, or kidney failure), signs or symptoms such as dyspnoea, low age-adjusted capillary oxygen saturation (SaO_2_) on room air and COVID-19 SEIMC score for each patient [17], and laboratory tests, including white cell count (SysmexXN-10), neutrophil-to-lymphocyte ratio, creatinine (mg/dL), urea (mg/dL), Na (mEq/L), pre-albumin (mg/dL) (Atellica-Siemens, Erlangen, Germany), C-reactive protein (CRP, mg/L) (Dimension EXL200S), blood urea nitrogen (*BUN*) $BUN=\frac{\mathrm{Urea}\;\left(\frac{\mathrm{mg}}{\mathrm{dL}}\right)}{2.1428}$, *BUN*/creatinine ratio (an emerging marker of venous congestion and poor prognosis, whose median population value is 15.0[IQR]:12.9–17.6) [18] and CRP/pre-albumin ratio (new inflammatory index and a potential predictor of complications) [19].

### 2.4. Sample Size Calculation

We tested the hypothesis that BI-derived overhydration status indicators were independent predictors of 90 d mortality in a multivariate model. We calculated the sample size using the findings of Basso et al. [31], where the effect of overhydration on the mortality showed an OR of 2.64 in critical care patients with a mortality rate of 31.6% in the overhydration compared to the normal hydration status (21.1%) group. Thus, with an alpha error of 0.05, a power of 80%, and a loss rate of 10%, a minimum of 89 patients were needed to attain sufficient power. To account for anticipated mortality, we aimed to recruit 120 patients.

### 2.5. Statistical Analysis

Statistical analyses of the data were primarily performed using the SPSS program (version 22.0.0 Windows, SPSS-Iberica, Spain). We used descriptive statistics to characterize our cohort of patients. Baseline characteristics were expressed as median and interquartile range (IQR) for continuous variables and as proportions for categorical variables. Furthermore, we categorized hydration percentage into tertiles (T) as T1 [lower 33rd percentiles of hydration status (<73.5%)], T2 [33rd–66th percentiles of hydration status (73.6–79%)], T3 [more than 66th percentile of hydration status (≥79.1%)]. We compared our hydration percentage tertiles with either the ANOVA test or Friedman test according to their distribution. Continuous variables were compared with Student’s t-test or Mann–Whitney U test according to their distribution. Categorical variables were compared with the chi-squared (or Fisher’s exact test). We also analyzed the relationship using Pearson or Spearman correlations models according to normal distribution.

In multivariate analysis, Cox proportional hazards regression was used to assess the relationship between hydration parameters and mortality in COVID-19 patients. Hazard ratio (HR) and their 95% confidence intervals (CI) were calculated. We used three models to analyze hydration status. In the first model, we analyzed hydration percentage (TBW/FFM, %), in the second one we analyzed ECW/TBW, and in the third one we analyzed TBW/weight. HR for death was expressed per 10% increase in hydration percentage, and 0.1 increase in ECW/TBW or TBW/Weight. To prevent potential confounding factors, the results were adjusted for several covariates that were known as potential risk or protective factors for mortality: age (y, continuous); sex (male–female); BMI (kg/m^2^, continuous); history of diabetes mellitus (absence–diagnosis); high blood pressure (absence–diagnosis); dyslipidaemia (absence–diagnosis); heart disease (absence–diagnosis), kidney failure (absence–diagnosis). We constructed three adjusted models: Model 1: adjusted for sex, age, and BMI. Model 2: additionally adjusted for the previous diagnosis of type 2 diabetes mellitus (T2DM), high blood pressure, dyslipidaemia, heart disease, or kidney failure. Model 3: additionally adjusted for body cell mass index (BCMI). Model 4: additionally adjusted for CRP level. Model 5: additionally adjusted for *BUN*/creatinine. Statistical significance was set at *p* < 0.05. Therefore, regarding the relationship between COVID-19 mortality and BI-derived hydration we analyzed multivariate logistic regression model where the response variable was mortality and predictor variables were hydration percentage > 76.15% and ECW/TBW > 0.58. The model then was modified with other factors: age (y, continuous), sex (male–female), BMI (kg/m^2^, continuous).

We also analyzed the multivariate logistic regression model, where the response variable was mortality and predictor variables were hydration percentage > 76.15% and ECW/TBW > 0.58.

The Kaplan–Meier product-limit estimator at 90 d was used to calculate the cumulative probability of death, estimate survival, and evaluate the difference among the hydration tertiles. The Kaplan–Meier survival curves were compared using log-rank Mantel–Cox) test. The time of origin was the admission day. The event was defined as death and all cases were censored at their last observation. Differences were considered statistically significant with a *p*-value < 0.05.

Evaluation of the diagnostic performance of individual hydration parameters was based on the receiver operating characteristic (ROC) curve and the area under the curve (AUC). We estimated the accuracy of hydration percentage using AUC by constructing a plot of sensitivity versus 1-specifity. ROC curves were used to determine the optimal cut-off values. These optimal cut-off points for each hydration measurement (ECW/TBW) and prognostic markers (CRP/pre-albumin, *BUN*/creatinine, COVID-19 SEIMC score) were determined by the point of convergence for greatest sensitivity and specificity by Youden index.

We computed the positive-negative predictive values (*PPV*-*NPV*) on Biagram^®^ observing the individual point distribution discriminated by the lower line, using Bayes’ theorem: $PPV=\frac{\mathrm{The}\;\mathrm{number}\;\mathrm{of}\;\mathrm{true}\;\mathrm{positives}\;}{\mathrm{Number}\;\mathrm{of}\;\mathrm{true}\;\mathrm{positive}\;+\;\mathrm{Number}\;\mathrm{of}\;\mathrm{false}-\mathrm{positive}\;}$ and NPV=negatives.

## 3. Results

### 3.1. Global Results

A total of 127 patients were admitted to the area of hospital care for COVID-19 and enrolled in the present study. COVID-19 study participants were predominantly males (59.1%), and their median (IQR) age was 69-y (59–80). Among the patients, 18.1% required additional intensive care in the Intensive Care Unit (ICU). The median length of hospital stay was 15 d (12–27) for the general ward, and 47 d (25–60, *p* < 0.001) for ICU patients. After 90 d, 111 patients (87.4%) had been discharged alive and 16 (12.6%) had died.

Table 1 shows the hydration parameters, COVID-19 SEIMC score, and biochemical measurements of the COVID-19 admitted patients.

Patients who did not survive showed a clinical score of greater severity [18.5 points (12.8–21.5) vs. 6 points (4–13), *p* < 0.001]. Non-survivor patients showed 8.3% high and 91.7% very high category, while survivor patients showed 7.9% low risk, 32.7% moderate risk, 18.8% high risk, and 40.6% very high-risk category of COVID-19 SEIMC score.

Comparing survivors and non-survivor patients, laboratory test also showed a greater *BUN*/creatinine [31.7 (25.3–43.3) vs. 22.8 (17.4–30.1), *p* < 0.001] as a marker of venous congestion and CRP/pre-albumin [1.19 (0.24–3.22) vs. 0.09 (0.02–0.31), *p* = 0.002] indicative of increased inflammatory status.

### 3.2. Hydration Status and 90 d Mortality

Hydration status distributions of COVID-19 patients based on BI-derived hydration percentage (TBW/FFM, %) showed a 48.8% euhydrated, while up to 43.3% classified with overhydration and 7.9% dehydration patients. In non-survivor patients, severe overhydration was associated with 37.5%, moderate overhydration with 18.8%, and mild overhydration with 25% (Figure 1A).

The distribution of individual hydration assessment data points of the COVID-19 patients showed a pattern of broad distribution in areas from dehydration to severe overhydration in the Biagram^®^ nomogram (Figure 1B). Patients who died are grouped in the lower-left region of the graph, consistent with states of overhydration. Biagram^®^ yields 100% *NPV* or 100% no mortality for subjects placed above the lower line (Figure 1B). Non-survivors had a significantly higher percentage of hydration compared to survivors [85.2% (76.9–89.3) vs. 73.7% (73.2–82.1), *p* = 0.001] and ECW/TBW [0.67 (0.50–0.75) vs. 0.54 (0.48–0.61), *p* < 0.001], while TBW adjusted for height (*p* = 0.429) or weight (*p* = 0.085) was not different between the groups of patients (Figure 1, Table 1).

Stratification of hydration values reveals greater mortality in the highest percentage hydration tertile (T3 vs. T1; 23.8 vs. 2.4%, *p* = 0.012) and increased need for ICU admission (T3 vs. T1, 31 vs. 9.5%, *p* = 0.027). The highest hydration tertile also had significantly increased CRP, CRP/pre-albumin, and more elevated sodium plasma levels, without differences in creatinine, glomerular filtration, or *BUN*/creatinine. The COVID-19 SEIMC score in the very high-risk category is related to higher hydration status of 62.5% vs. 20.0% (T3 and T1 of hydration percentage) (Table 2).

The Kaplan–Meier plot was constructed using tertiles of the hydration distribution. Kaplan–Meier product-limit estimator showed that higher hydration percentage (T3) was significantly linked with higher mortality rates (log-rank test, *p* = 0.010) (Figure 2). Median of survival time was 79.2 d (68.8–89.6) in T3 (hydration percentage ≥ 79.1%), 89.4 d (82.3–96.5) for T2 (hydration percentage: 73.6–79%) and 95.7 d (93.3–98.2) for T1 tertile (hydration percentage ≤ 73.5%).

### 3.3. Optimal Hydration Parameters Cut-Off Value and 90 d Mortality Prediction in COVID-19 Disease

Using ROC curves, we determined the hydration status measurement cut-off points for predicting mortality. A 76.15% hydration percentage value was the most sensitive (81.3%) and specific (64%) factor, with an AUC was 0.746, for mortality risk prognosis in acute COVID-19. For ECW/TBW, AUC was 0.841, and the 0.58 value was the most sensitive (93.8%) and specific (67.6%) factor for mortality risk prognostic in acute SARS-CoV-2 infection. We compared these data with the cut-off points of mortality prediction using ROC curves for the analytical parameters established as prognostic factors (Figure 3).

COVID19 SEIMC score greater than 10.5 points, CRP/pre-albumin greater than 0.88, and *BUN*/creatinine greater than 23.98 were also the cut-off points for predicting mortality in COVID-19 patients. ECW/TBW showed the highest sensitivity concerning the rest of the parameters in predicting mortality at 90 d, while CRP/pre-albumin was the highest specificity (Table 3).

Hydration percentage and ECW/TBW showed a positive correlation with COVID-19 clinical score (r = 0.394, *p* < 0.001 and r = 0.541, *p* < 0.001, respectively), CRP (r = 0.220, *p* = 0.013 and r = 0.283, *p* = 0.001, respectively), CRP/pre-albumin (r = 0.263, *p* = 0.026 and r = 0.379, *p* = 0.001, respectively) and *BUN*/creatinine (r = 0.218, *p* = 0.015 and r = 0.281, *p* = 0.001, respectively). ECW/TBW also showed a positive correlation with hospital stay in survivors of COVID-19 (r = 0.310, *p* = 0.001).

In the assessment of hydration status to predict mortality in COVID-19 patients, we performed multivariate logistic regression models with mortality as a response variable: in the first model, we evaluated the association between mortality and the presence of overhydration percentage ≥76.15% [OR 7.692; 95%CI (2.067–28.616); *p* = 0.002] and the second model the predictor variable tested was ECW/TBW ≥ 0.58 [OR 30.000; 95%CI (3.815–235.933); *p* = 0.001]. We found a significant association with mortality in both models of overhydration, which is maintained even after adjusting for confounding variables such as age, sex, and BMI, [OR 5.804; 95%CI (1.460–23.071); *p* = 0.013] and [OR 18.237; 95%CI (2.196–151.457); *p* = 0.007], respectively.

We used an 11-component model multivariate analysis (by Cox regression) to evaluate the utility of the hydration status measurements as prognostic indicators of mortality in COVID-19. We found that an increased hydration percentage value was significantly associated with a higher mortality hazards ratio [HR 2.967; 95%CI (1.459–6.032), *p* = 0.003). This trend was also maintained in the adjusted models by the confounding variables (Table 4).

Likewise, the ECW/TBW [HR 2.528; 95%CI (1.664–3.843), *p* < 0.001] was also associated with an increase in mortality HR of the crude model, with this relationship maintained in the adjusted models. These results are significantly greater than traditional clinical measures of fluid overload such as CRP/prealbumin [HR 3.057; 95% CI (0.906–10.308), *p* = 0.072) or *BUN*/creatinine [HR 1.861; 95% CI (1.375–2.520), *p* < 0.001)].

## 4. Discussion

Overhydration is a clinical finding among COVID-19 patients. Reports indicate that patients with acute respiratory distress syndrome (ARDS), to which COVID-19 frequently progresses, develop excessive fluid accumulation. Cumulative net fluid balance was significantly greater among non-survivors, compared to survivors, of ARDS during a 30 d period [32]. COVID-19 patients also can present with acral [3] and pulmonary edema [4]. These observations stimulated research interest in hydration as a prognostic factor for morbidity and mortality in COVID-19.

The main finding of this study was a positive association between overhydration, assessed by BIVA, and mortality in COVID-19 patients, determining that BI-derived hydration parameters were significant 90- mortality predictors. In the Kaplan–Meyer plot of the survival curve, a hydration percentage of >79.1% (highest hydration tertile, T3) grouped most deceased patients, as a significant mortality predictor. Importantly, the present findings provide the first evidence that BIVA-related estimates of fluid overload were significantly better predictors than standard biochemical indicators of overhydration and other clinical scores. Thus, to our knowledge, this is the first report of the deleterious relationship between hydration status, COVID-19, and mortality.

Beyond these data, the survival analysis revealed an increase in hydration percentage, and ECW/TBW have more prognostic value than overhydration indicators associated with inflammation (CRP/prealbumin) or vascular overload (*BUN*/creatinine).

There are two fundamental lines in the pathogenesis of increased hydration, the inflammatory component of the disease and primary fluid retention due to cardiac or renal hemodynamic failure.

In more than a third of the patients in our sample, the mean hydration of FFM percentage was estimated to be 86.6% (84.1–90) and ECW/TBW was 0.67(0.61–0.74), showing a severe situation of inflammation/fluid retention associated with SARS-CoV-2 infection. In critical patients, the global effect of overhydration on mortality revealed its negative effect with OR 8.16 [1.55–6.00] [8]. Samoni et al. [33] in the 125 critical patients cohort, described 64.8% BI-derived overhydration status with a mean hydration percentage of 80.68 ± 5.82%, while in our series we found an overhydration percentage slightly lower. The higher incidence of sepsis and multiple organ failure in critical patients constitutes inflammatory settings favorable to more fluid retention.

In the present study, patients with fluid overload showed significantly elevated levels of inflammatory parameters, CRP, and CRP/prealbumin. These findings indicate that overhydration status in COVID-19 admitted patients show a poorer outcome related to the inflammatory pattern. In ICU patients, Li et al. [34] reported that an increase in CRP/prealbumin independently correlated with worse clinical outcomes. CRP/prealbumin ROC curve in predicting hospital mortality was 0.701 (0.597–0.805), *p* = 0.001. In our cohort, the ROC curve was similar for COVID-19 patients. To the best of our knowledge, the current study is the first time to demonstrate that CRP/prealbumin is an independent predictor of mortality in admitted COVID-19 patients. Furthermore, the prognostic value of the CRP/prealbumin for predicting mortality in COVID-19 is significantly greater than the traditional risk factor, CRP [34].

COVID-19 patients with pneumonia require meticulous fluid management to avoid circulatory overload. Although fluid administration is important in patients with shock, excessive fluid administration in critical patients is associated with poor outcomes [33]. Based on these results, we must consider that conservative fluid management should be employed among COVID-19 patients when no evidence of shock is present. Particularly, patients with ARDS are more likely to be harmed from unnecessary fluid administration.

In the absence of specific data on the role of hyperhydration in inflammatory processes such as COVID-19, the heart failure model may be useful to see the effect of hydration on clinical outcomes. In heart failure, fluid retention caused an increased risk of mortality [OR 3.05 (1.55–6.00)] [8]. Hydration percentage discriminated (*p* = 0.001) surviving [74.00% (73.48–81.15)] and non-surviving [85.00% (74.02–87.95)] in these patients [35], we also found similar differences between survivor and non-survivor patients.

We also found significant differences in *BUN*/creatinine, a congestion vascular marker, between surviving and non-surviving patients. A recent study revealed that *BUN*/creatinine may be associated with disease severity in COVID-19 with optimal thresholds of the *BUN*/creatinine at 51.7 had a superior possibility for mortality [AUC 0.950, (90% sensibility, 92% specificity), log-rank test, *p* < 0.001] [36].

The predictive value of the hydration percentage depends on the choice of cut-off points, which should be developed for the specific pathology [8]. Using ROC analysis, Santarelli et al. [35] demonstrated a similar probability of 90 d cardiovascular death [AUC 0.715; 95% CI(0.65–0.76), *p* < 0.04] predicted by using BIVA-derived hydration percentage. Similarly, in our study, the cut-off value of 76.15% hydration provides a clear prognostic factor with a very high risk of mortality [OR 7.692; 95%CI (2.067–28.616); *p* = 0.002]. Furthermore, ECW/TBW shows a higher diagnostic potency than that found in other pathologies. In chronic kidney disease, it was found an AUC = 0.64 (cut-off: 0.40, 61% sensitivity, 60% specificity, *p* = 0.002) [37]. Consequently, our results confirm a strong association between hydration status and survival in COVID-19 patients, which is more useful to use our cut-off point (0.58) since it has shown greater specificity and sensitivity in this pathology. Using 0.58 as a cut-off point, our results could more clearly establish an objective threshold for discriminating prognosis of mortality [OR 30.000; 95%CI (3.815–235.933); *p* = 0.001]. The cut-off value conventionally established in the literature (0.40) reaches 100% sensitivity but only 4.5% specificity.

A recent clinical COVID-19-SEIMC score [17] has been validated to establish the severity prognosis of COVID-19. In our series, we found a similar relationship between score and mortality with equivalent sensitivity and specificity values to overhydration parameters. Overhydration parameters have proven to be efficient as the COVID-19 score for this purpose in our series, an easy measure could replace a multidimensional score [10].

## 5. Conclusions

Our results identify levels of overhydration using BIVA associated with increased COVID-19 severity, 90 d mortality risk, and a hydration tertiles distribution related to the median survival time. We further demonstrate the increased prognostic value of BIVA hydration assessments compared to traditional clinical indicators of hydration.

Given its prognostic impact, these findings suggest that the evaluation of hydration status by BIA variables is adequate for bedside evaluation. It could be routinely included in COVID-19 clinical assessment at hospital admission, allowing adequate intervention to reduce mortality risk. This is recommended to maintain a restrictive fluid strategy in these patients. Therefore, careful evaluation and treatment of overhydration status are essential in COVID-19 patients, in favor of an individualized approach for restrictive fluid management providing more attention to the use of dynamic methods for evaluation of fluid change.

Whereas our findings of BI assessments are provocative, further studies are needed to validate the use of BIA in COVID-19 hydration status assessment and to incorporate this tool combined with clinical and biochemical parameters into hospitalization medical protocols.

## 6. Strengths–Limitations

We highlight, as strengths of our study, that is the first study that evaluated the association of overhydration—using BIVA—with mortality in COVID-19 hospitalized patients. Different BIA parameters may be useful for the early diagnosis of patients with fluid retention, allowing the provision of adequate treatment to reduce mortality risk. Overhydration assessment could be considered as an additional parameter in the clinical management of COVID-19 patients and patients with situations of acute inflammation.

The limitation in our study is the use of only admission BI assessments of hydration. There is a need for studies of serial BI measurements to monitor intra-individual variation due to COVID-induced fluid changes and the effects of clinical interventions on hydration, in order to assist in medical care and changing treatment to benefit patients.

## Figures and Tables

**Figure 1 nutrients-14-02726-f001:**
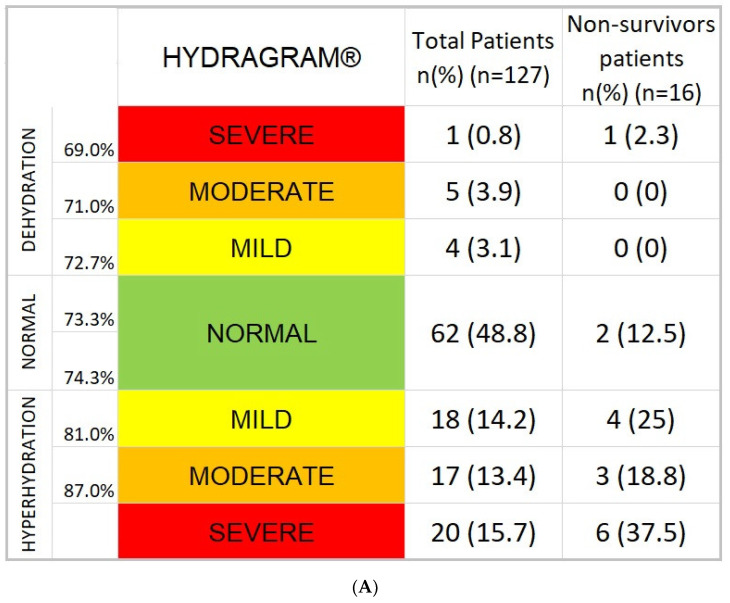
(**A**) Hydragram^®^ graph of COVID-19 disease (*n* = 127). Hydration status distribution of COVID-19 admitted patients showed a situation of hyperhydration associated with COVID-19 (43.3%). (**B**) Biagram^®^ graph of COVID-19 disease (*n* = 127): distribution of COVID-19 admitted patients. All casualties are marked with red dots. Blue dots below lower line are false positive, blue dots above lower line true negative. The position of the points is related to the E/I status of the patient based on raw measurements of reactance and phase angle. Points below the lower line are associated with states of expansion of extracellular space greater than intracellular (severe inflammatory process), between the line points are normal E/I and above the upper line extracellular is smaller than intracellular space. The entire cluster of red dots is positioned below the lower discriminating line showing above the line a 100% *NPV* (negative = survivors) area.

**Figure 2 nutrients-14-02726-f002:**
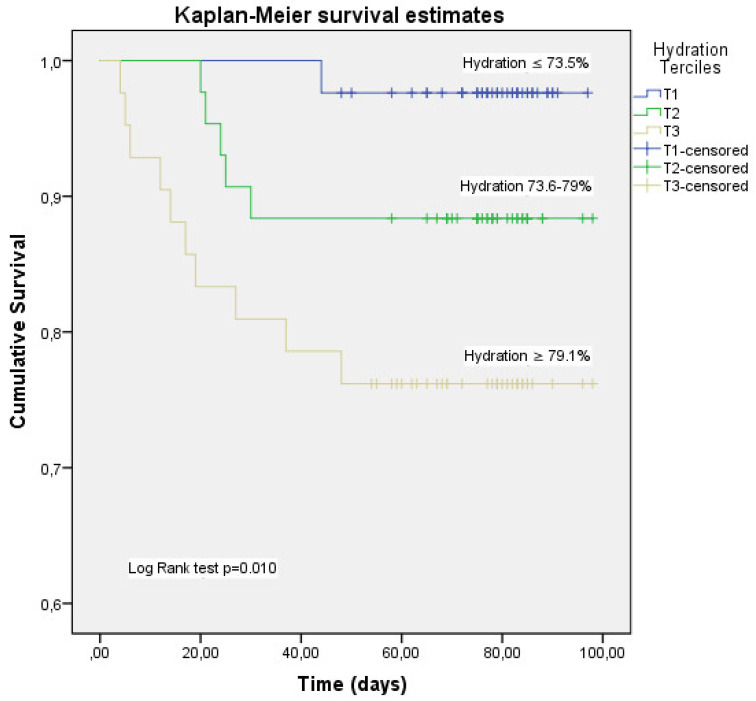
Kaplan–Meier analysis for the cumulative percentage of surviving patients at 90 days according to hydration percentage tertiles. The 33rd and 66th percentiles of hydration percentage were used as the cut-off point to divide the patients with acute COVID-19 disease into 3 groups (T1, T2, T3) and made the Kaplan–Meier plot. Kaplan–Meier product-limit estimator shower that higher hydration (T3) was significantly linked with higher mortality rates (log-rank test, *p* = 0.010). Mortality was mainly concentrated in T3.

**Figure 3 nutrients-14-02726-f003:**
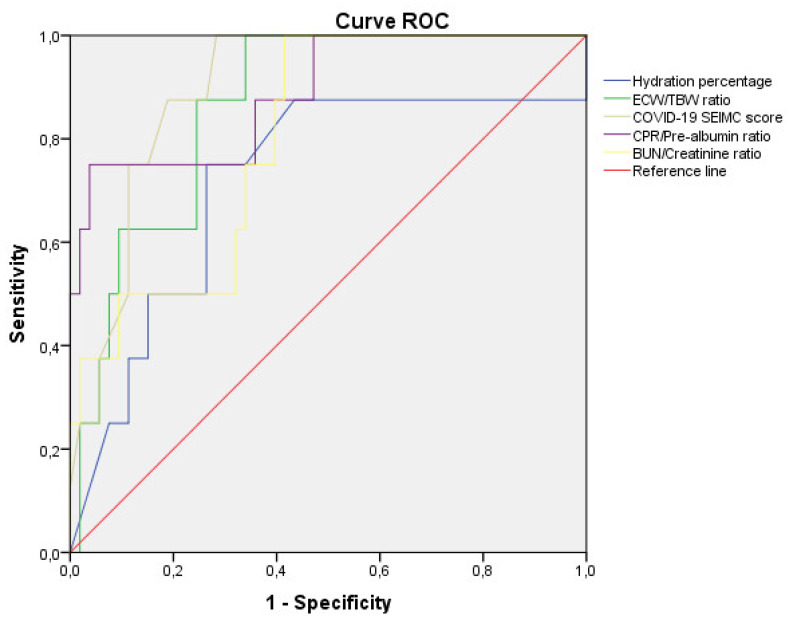
Comparative analysis of ROC curve of hydration with established prognosis factor in COVID-19 patients. Comparative analysis of the receiver operating characteristic curves of hydration percentage, ECW/TBW ratio, COVID-19 SEIMC clinical score, and analytical indicators (CRP/pre-albumin, *BUN*/creatinine) for prediction of mortality in patients with COVID-19 (*n* = 127).

**Table 1 nutrients-14-02726-t001:** Physical characteristics and biochemical measurements of COVID-19 patients related to survival and mortality.

	COVID-19 Patients	COVID-19 Survivors	COVID-19 Non-Survivors	
	Median (IQR)	Median (IQR)	Median (IQR)	*p* ^a^
	(*n* = 127)	(*n* = 111)	(*n* = 16)	
Age (years)	69 (59–80)	68 (56–77)	84 (70–88)	0.001
Male, *n* (%)	75 (59.1)	66 (59.5)	9 (56.3)	0.807
Hydration (TBW/FFM, %)	73.8 (73.3–84.3)	73.7 (73.2–82.1)	85.2 (76.9–89.3)	0.001
ECW/TBW	0.55 (0.49–0.63)	0.54 (0.48–0.61)	0.67 (0.59–0.75)	<0.001
TBW/H (L/m)	0.24 (0.21–0.26)	0.24 (0.21–0.26)	0.22 (0.20–0.25)	0.429
TBW/body weight (%)	52.4 (48.1–56.1)	51.9 (48.1–55.9)	56 (47–63.3)	0.085
COVID-19 SEIMC Score	7 (4–15.5)	6 (4–13)	18.5 (12.8–21.5)	<0.001
Low risk category *n* (%)	7.1	8 (7.9)	0 (0)	0.333
Moderate risk category *n* (%)	29.2	33 (32.7)	0 (0)	0.019
High risk category *n* (%)	17.7	19 (18.8)	1 (8.3)	0.369
Very high risk category *n* (%)	46	41 (40.6)	11 (91.7)	0.001
Creatinine (mg/dL)	0.85 (0.71–1.04)	0.82 (0.71–1.01)	1.07 (0.71–1.39)	0.04
GF (mL/min/1.73 m^2^)	81 (64.75–90)	82 (70–90)	55 (45.3–82.3)	0.004
Na (mEq/L)	141 (139–144)	141 (139–144)	144.5 (139.3–147)	0.046
CRP (mg/L)	16.7 (5.0–59.9)	14.3 (4.2–44.7)	97.5 (24.4–199.6)	<0.001
CRP/Pre-albumin	0.26 (0.10–0.67)	0.25 (0.08–0.37)	1.06 (0.35–1.23)	0.002
*BUN*/Creatinine	24.2 (18.7–31.1)	22.8 (17.4–30.1)	31.7 (25.3–43.3)	0.001

^a^ *p* for comparison of non-survivors and survivors. IQR: interquartile range; TBW: total body water, H: height; FFM: fat free mass; ECW: extracellular water; SEIMC: Spanish Society of Infectious Diseases and Clinical Microbiology, GF: Glomerular filtration; Na: sodium; CRP: C-reactive protein; *BUN*: Blood urea nitrogen.

**Table 2 nutrients-14-02726-t002:** Characteristics of COVID-19 patients according to tertiles of hydration status (percentage TBW/FFM).

Variables	(T1) ≤73.5	(T2) 73.6–79	(T3) ≥79.1	*p*
	(*n* = 42)	(*n* = 43)	(*n* = 42)	
	Median (IQR)	Median (IQR)	Median (IQR)	
Mortality ratio (%)	1/16 (2.4)	5/16 (11.6)	10/16 (23.8)	0.012
ICU admission ratio (%)	4/23 (9.5)	6/23 (14)	13/23 (31)	0.027
Age (y)	63 (54–71)	73 (61–83)	71 (62–85)	<0.001
Male sex *n* (%)	28 (66.7)	23 (53.5)	24 (57.1)	0.445
BMI (kg/m^2^)	25.6 (23.2–29.3)	27.3 (24.2–30.9)	26.2 (24.2–30.6)	0.374
Hydration (%): TBW/FFM	73.1 (72.7–73.3)	73.8 (73.7–76.3)	86.6 (84.1–90)	<0.001
ECW/TBW	0.46 (0.43–0.49)	0.55 (0.52–0.59)	0.67 (0.61–0.74)	<0.001
TBW/H (L/m)	0.22 (0.19–0.26)	0.23 (0.20–0.25)	0.25 (0.23–0.30)	<0.001
TBW/body weight (%)	51.5 (48.6–55.1)	50.8 (45.4–54.1)	57.2 (51.4–62.8)	<0.001
COVID-19 SEIMC Score	5 (3–7.5)	10 (5–16)	11.5 (6–19)	0.003
Low risk category *n* (%)	4 (10.0)	2 (4.9)	2 (6.3)	0.66
Moderate risk category *n* (%)	18 (45.0)	11 (26.8)	4 (12.5)	0.01
High risk category *n* (%)	10 (25.0)	4 (9.8)	6 (18.8)	0.196
Very high risk category *n* (%)	8 (20.0)	24 (58.5)	20 (62.5)	<0.001
Creatinine (mg/dL)	0.85 (0.73–0.99)	0.81 (0.71–1.03)	0.85 (0.65–1.11)	0.964
GF (mL/min/1.73 m^2^)	85 (74.5–90)	80 (64–90)	76 (59–90)	0.132
Na (mEq/L)	140 (139–142)	141 (138–144)	144 (140–146)	0.009
CRP (mg/L)	12.9 (4.6–55.9)	13 (4–31.4)	30.2 (11.8–148.5)	0.019
CRP/Pre-albumin	0.255 (0.028–0.372)	0.125 (0.018–0.213)	0.440 (0.220–1.215)	0.011
*BUN*/Creatinine	21.3 (18.1–29.3)	24.6 (17.7–31.7)	26.0 (19.3–37.4)	0.14

IQR: interquartile range; ICU: Intensive Care Unit; BMI body mass index; TBW: total body water, FFM: fat free mass; ECW: extracellular water; H: height; SEIMC: Spanish Society of Infectious Diseases and Clinical Microbiology, GF: Glomerular filtration; Na: sodium; CRP: C-reactive protein; *BUN*: Blood urea nitrogen.

**Table 3 nutrients-14-02726-t003:** Analysis of the prognostic factors of mortality in COVID-19.

Variables	AUC	95% CI Lower-Upper Bound	Cut-Off Point	Sensitivity	Specificity	*p*
Hydration (%)	0.746	0.618–874	76.15%	81.30%	64%	0.002
ECW/TBW ratio	0.841	0.762–0.920	0.58	93.80%	67.60%	<0.001
COVID-19 SEIMC Score	0.858	0.770–0.947	10.5	91.70%	68.30%	<0.001
CRP/Pre-albumin ratio	0.794	0.616–0.973	0.88	72.70%	90.20%	0.002
*BUN*/Creatinine ratio	0.77	0.659–0.880	23.98	93.80%	54.10%	0.001

AUC: area under the curve; TBW: total body water, ECW: extracellular water; SEIMC: Spanish Society of Infectious Diseases and Clinical Microbiology, CRP: C-reactive protein. Evaluation of prognosis factors of mortality in COVID-19 based on the area under the curve (AUC) of the receiver operating characteristic (ROC) curve, and sensitivity, and specificity values to determine the optimal cut-off values.

**Table 4 nutrients-14-02726-t004:** Multivariate Cox regression for hydration status as predictors of mortality in COVID-19 patients.

	Hydration Percentage		ECW/TBW Ratio		TBW/Weight Ratio	
	HR (CI)	*p*	HR (CI)	*p*	HR (CI)	*p*
Crude model	2.967 (1.459–6.032)	0.003	2.528 (1.664–3.843)	<0.001	1.059 (1.001–1.120)	0.046
Model 1	2.514 (1.281–5.236)	0.014	2.277 (1.355–3.828)	0.002	1.024 (0.957–1.096)	0.492
Model 2	2.849 (1.259–6.448)	0.012	2.692 (1.496–4.846)	0.001	1.027 (0.956–1.102)	0.469
Model 3	2.933 (1.215–7.077)	0.017	3.043 (1.582–5.851)	0.001	1.025 (0.956–1.100)	0.485
Model 4	2.384 (1.067–5.328)	0.034	2.449 (1.236–4.854)	0.010	1.036 (0.968–1.109)	0.308
Model 5	2.365 (0.942–5.937)	0.067	2.537 (1.203–5.352)	0.014	0.996 (0.921–1.078)	0.921

The hazard ratio (HR) for death was expressed per 10% in hydration percentage and 0.1 in ECW/TBW or TBW/weight ratio, for a univariate model and sequential adjustment models. Dependent variable: survivors (0) vs. non-survivors (1). Cox regression was expressed by HR and 95% confidence interval (CI). Model 1: adjusted for sex, age, and BMI. Model 2: additionally, adjusted for diagnosis previous of type 2 diabetes, high blood pressure, dyslipidaemia, heart disease, and kidney failure. Model 3: additionally, adjusted for malnourished patients, defined as BCMI less than 10.5 in men and 7.5 in women. Model 4: additionally, adjusted for CRP level. Model 5: additionally, adjusted for *BUN*/Creatinine.

## Data Availability

The datasets generated during and/or analyzed during the current study are available from the corresponding author on reasonable request.

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
