# Peer review of "Overhydration Assessed Using Bioelectrical Impedance Vector Analysis Adversely Affects 90-Day Clinical Outcome among SARS-CoV2 Patients: A New Approach"

_nutrients, 2022, doi:10.3390/nu14132726_

Round 1
Reviewer 1 Report
The authors concluded that BI using BIVA is useful for estimating clinical outcome among SARS-CoV2 patients. I think that this manuscript is well compiled, concise, and very well written. But some issues should be resolved.
1. The authors performed multivariate logistic regression analysis in 5 models. But the model including COVID-19 SEIMC Score was not performed. Does hydration percentage or ECW/TBW still have a significant predictive value for mortality in the model including COVID-19 SEIMC Score?
2. Bioimpedance spectroscopy (BIS) is also widely used in clinical practice. Although I don’t think that BIS should be used, is there any reason why BIVA was used instead of BIS?
3. There are significant differences in ECW/TBW between the lean and obese subjects (Chamney et al. A whole-body model to distinguish excess fluid from the hydration of major body tissues. Am J Clin Nutr 2007;85:80-89). Although age and sex were adjusted for ECW/TBW, how is the problem between lean and obese subjects resolved in this manuscript?
4. How are age, sex, and obesity issues adjusted for TBW/FFM?
5. As hydration percentage increases, Na concentration also increases. How would the authors explain the reason for this phenomenon?
6. Table 1 is incomplete with two blanks.
7. Are laboratory data also measured within 72h after hospital admission?
8. Even when some abbreviations appear first, they are written as abbreviations as follows: BI, line 25 abstract; BI, line 73; BIA, line 110; FFM, 115; E/I, line 130; T2D, line 188; BCMI, line 190; med, line 446
9. CRP is incorrectly written as CPR in some points (line 190, line 355)
Author Response
Dear Editor and Reviewers,
We would like to thank you very much for your constructive comments and suggestions which have undoubtedly helped us to improve our manuscript.
We have taken these comments and suggestions into consideration and have revised the paper accordingly. We have made all possible efforts to respond to each of the reviewers’ comments and have edited the manuscript where we were able to fully address the reviewers’ suggestions.
We have provided the replies to the comments in the following section and have highlighted changes in the manuscript in red font.
We hope that our revised manuscript may now be found acceptable for publication in the journal. Nevertheless, we are of course willing to revise it further according to any other suggestions or concerns raised by the Editor or the Reviewers.
Yours faithfully,
Isabel Cornejo-Pareja,

Reviewer 2 Report
This manuscript estimates the body water content of SARS-Cov2 patients by the bioelectrical impedance method, and attempts to predict the prognosis by the water balance.
This study is challenging and fits into the "Hydration" section.
However, it is not a highly planned epidemiological study,because it is a pandemic study.
It is understandable that there are more limitations, but it is necessary to express efforts to reduce limitations, for clinical application.
・period from onset to hospitalization
・when using BIA, break time before using BIA
・intra-individual variation for this patient(s)
Please describe a mini-review of COVID19 prognostic factors.
And clarify the position of this study.
Please explain why the authors use the logistic regression model. The time effect might be needed.
I recommend Inverse Probability Weighting or another adjustment model for survival analysis.
Please consider the possibility that using bootstrap method for ROC analysis.
There are several methods to calculate the cutoff value using the ROC curve, please cited.
Please explain why using 90-day follow-up data. In the clinical setting, it might be significant the average days of mortality or recovery.
In the determinant cutoff value using ROC analysis, is sensitivity analysis unnecessary. For example, age, sex.
Please consider the information to add to each section of introduction, discussion, and strength-limitations.
Please describe the abbreviations which appear for the first time in full form.
Author Response

(The authors gave the same response as above.)
